# The Impact of Genital Ulcers on HIV Transmission Has Been Underestimated—A Critical Review

**DOI:** 10.3390/v14030538

**Published:** 2022-03-05

**Authors:** João Dinis Sousa, Viktor Müller, Anne-Mieke Vandamme

**Affiliations:** 1Laboratory of Clinical and Epidemiological Virology, Department of Microbiology, Immunology and Transplantation, Rega Institute for Medical Research, KU Leuven, B-3000 Leuven, Belgium; annemie.vandamme@kuleuven.be; 2Global Health and Tropical Medicine, Instituto de Higiene e Medicina Tropical, Universidade Nova de Lisboa, 1349-008 Lisbon, Portugal; 3Institute of Biology, Eötvös Loránd University, 1117 Budapest, Hungary; mueller.viktor@ttk.elte.hu; 4Institute for the Future, KU Leuven, B-3000 Leuven, Belgium

**Keywords:** HIV, HIV-1, genital ulcer disease, sexually transmitted infection, HIV transmission, co-factor of HIV transmission, per-act transmission probability

## Abstract

In the early 1990s, several observational studies determined that genital ulcer disease (GUD), in either the index or the exposed person, facilitates HIV transmission. Several meta-analyses have since presented associated risk ratios (RR) over the baseline per-act transmission probability (PATP) usually in the range of 2–5. Here we review all relevant observational studies and meta-analyses, and show that the estimation of RRs was, in most cases, biased by assuming the presence of GUD at any time during long follow-up periods, while active genital ulcers were present in a small proportion of the time. Only two studies measured the GUD co-factor effect in PATPs focusing on acts in which ulcers were present, and both found much higher RRs (in the range 11–112). We demonstrate that these high RRs can be reconciled with the studies on which currently accepted low RRs were based, if the calculations are restricted to the actual GUD episodes. Our results indicate that the effect of genital ulcers on the PATP of HIV might be much greater than currently accepted. We conclude that the medical community should work on the assumption that HIV risk is very high during active genital ulcers.

## 1. Introduction

It is widely accepted that genital ulcer disease (GUD) caused by various sexually transmitted infections (STI) facilitates sexual transmission of human immunodeficiency viruses (HIV). It may cause bleeding, and it causes local inflammation and an upregulation of CCR5 expression in T cells, thus favouring local HIV replication, and increasing the odds of transmission from the index partner. In the exposed partner, GUD provides a portal of entry and increases the number of target cells available locally for HIV [1]. Treatment of GUD is an accepted measure to reduce HIV transmission [2,3,4].

STIs and GUDs appear to be more important determinants of HIV transmission in Africa when HIV epidemics are not yet mature, fueled by high-risk groups [4,5,6]. Their importance was probably very high in the 1970s and 1980s, when the HIV pandemic was expanding fast, and subsequently subsided, as HIV became generalized. With the current UNAIDS 95-95-95 targets (95% diagnosed among all people living with HIV, 95% on antiretroviral therapy (ART) among diagnosed, and 95% virally suppressed), it is expected that the epidemic will again become concentrated in disadvantaged and high-risk groups. This will again increase the importance of STIs and GUDs for HIV epidemics [4,6].

Focusing on heterosexual relations in low-income countries, several observational studies attempted to measure how much GUD increases the per-sexual act transmission probability (PATP) of HIV-1 [7,8,9,10,11,12,13] and presented the results in the form of risk ratios. These studies have been the subject of several systematic reviews and meta-analyses [1,14,15,16,17]. These reviews have described central estimates of GUD-related risk ratios for HIV PATP of 2.2–11.3 [1], 1.7–3.1 (for Herpes simplex virus type 2 (HSV-2) seropositivity) [14], 5.3 [16] and 2.65 [17].

However, certain studies determined much higher risk ratios. In a landmark study, Cameron et al., 1989 [7] selected Kenyan male subjects presenting with a non-HIV STI (often involving GUD, as well) who reported a single recent contact with a commercial sex worker (CSW), and who did not have additional CSW contacts in subsequent weeks. The authors tracked HIV seroconversions in this study population, and estimated PATPs under the assumption that HIV infections resulted from these single exposures. The PATP for men presenting with GUD was 16.22% (3.24–29.2). For uncircumcised men with GUD, the authors calculated a PATP of 42.8% (12.7–73.0), using survivorship analysis [7]. Most observed GUDs were chancroid caused by the bacterium *Haemophilus ducreyi* [7], which is almost always acquired by men from CSWs. Based on their study design, the authors suggest that each GUD-presenting male acquired both GUD and HIV in the same single CSW contact [7]. This implies that they were measuring the effect that GUD in index persons had on HIV transmission (i.e., an index_GUD_→exposed situation, rather than an index→exposed_GUD_ one). The measured effect of GUD was extremely high compared with all other studies. Was this an outlier result, fundamentally incompatible with other measurements, or can we reconcile them?

## 2. Materials and Methods

### 2.1. Calculating Diluted and Undiluted Risk Ratios

In an observational study, if no other risk factor is present, the PATP, P, can be calculated from the formula:(1)$$I=1-{\left(1-P\right)}^{n}$$
where *I* is the observed incidence of HIV in initially seronegative people during the entire follow-up period (or in a subperiod of it) and n is the average number of sexual exposures to HIV-positive partners during that period. *P* is the baseline PATP observed when no facilitating co-factor such as GUD or acute infection is present. Many different studies estimated it for heterosexual relations, and meta-analyses show that it is about 0.05–0.1% in high-income countries and 0.1–0.4% in low-income countries [14,15,16]. If GUD is present in some exposures only, the per-act risk in these exposures can be denoted by *P_g_*, the risk ratio being *R_g_* = *P_g_*/*P*. As described in Hayes et al., 1995 [8], in an observational study, the cumulative risk or incidence, *I_g_*, is given by:(2)$$I_{g}=1-{\left(1-P\right)}^{n_{0}}{\left(1-P_{g}\right)}^{n_{g}}$$
where *n*_0_ and *n_g_* are the number of sexual exposures to HIV-positive partners without and with the risk factor, respectively [8]. Given the other parameters, the PATP for acts involving the additional risk factor is:(3)Pg=1−[1−Ig(1−P)n0]1ng

Equations (2) and (3) can be applied to the effect of genital ulcers present in both the index person (index_GUD_→exposed) or in the exposed person (index→exposed_GUD_). However, if the co-factor is assumed to have been present in *all* exposures, it is possible to use a formula similar to Equation (1):(4)$$I_{g}=1-{\left(1-P_{gdil}\right)}^{n}$$

This permits to calculate *P_gdil_*, the PATP for participants who had the co-factor, and, observing a cohort of people who did not have the co-factor, to use Equation (1) to calculate P, and a crude formula for the risk ratio is *R_gdil_* = *P_gdil_*/*P*. Returning to the case of GUD, most researchers assigned “GUD status” to participants who reported genital ulcers *at any time* during the follow-up period or subperiod. They measured *I_g_* for these participants and then calculated *P_gdil_* by Equation (4), assuming that the GUD co-factor was constant over the follow-up period, regardless of whether active genital ulcers were present. However, follow-up subperiods are usually several months to one year, and in most GUDs, active genital ulcers only last about one to a few weeks. Thus, the resultant *P_gdil_* falls between *P* and *P_g_*, thus underestimating the latter. It is a *diluted* measure of the GUD-associated PATP, and hence we call it *P_gdil_*. Both *P_gdil_* and the corresponding risk ratio, *R_gdil_* will be lower, and probably *much lower*, than the *P_g_* and *R_g_* computed by the method that uses Equation (3) (the *undiluted* PATP and risk ratio, respectively).

### 2.2. Retrieval of PATP Estimates

We were interested in reviewing all primary research articles that measured the effects of GUD in HIV PATPs. As a first step, we obtained all the systematic reviews and meta-analyses of measurements of HIV PATPs. The last published meta-analysis was Boily et al., 2009 [16]. We retrieved all primary articles referenced or cited in this and the other meta-analyses.

We then extended our efforts to studies published after 2008. We performed searches in Google Scholar that, together, were equivalent to the following formula:

“HIV” AND “Africa” AND “follow-up” AND (“transmission probability” OR (“probability of transmission”) AND (“per act” OR “per contact” OR “per sexual act”) AND (year ≥ 2009)

For all primary papers the first author (JDS) read the abstracts and determined whether the study was empirical and if either incidence or transmission probability of HIV were studied. If these conditions were met, the full text was assessed.

For all primary papers we determined whether the GUD-associated PATPs and/or risk ratios were diluted or undiluted—i.e., whether they were measuring *P_gdil_* or *P_g_*, or *R_gdil_* or *R_g_*, using the terminology of Section 2.

## 3. Results

### 3.1. The Studies That Estimated GUD Effects on HIV-1 PATPs

The meta-analysis of PATPs published by Boily et al., 2009 [16] is a comprehensive and influential review of HIV-1 PATPs, and associated co-factors. Among many studies reviewed, they list all studies known to that date that estimated the effects of GUD (either in the index or the exposed person) in HIV PATPs [7,8,9,10,11,12,13]. A subsequent systematic review by Patel et al., 2014 [17] only identified one additional study [18]. A meta-analysis published by Looker et al., 2017 [19] reviewed many studies measuring or estimating the effects of Herpes simplex virus type 2 (HSV-2) infection on HIV transmission, but did not focus on PATPs. We reviewed all the studies listed in it that were not yet covered by the previous meta-analyses and none of them attempted to calculate HSV-2 or GUD effects on HIV PATPs.

We searched the post-2008 literature and found many post-2008 studies estimating co-factor effects on HIV incidence—a non-exhaustive list of representative studies is: [20,21,22]. However, we found only two studies that estimated PATPs of HIV considering also the effects of GUD in them [18,23]. One of them [18] was included in the review by Patel et al. [17]. The other one [23] calculated risk ratios of HSV-2 for PATP of HIV, but pooled together HSV-2 in the index and in the exposed, and therefore we rejected it. We found no meta-analysis on the GUD co-factor effect on HIV PATPs published after the paper by Boily et al., 2009 [16]. We thus built Table 1 with all the relevant studies, the ones already covered by the Boily et al., 2009 meta-analysis, and the additional one [18]. The table is likely to contain all or almost all relevant studies of the last three decades.

As Table 1 shows, only two studies [7,8] calculated undiluted PATPs. In the Cameron et al., 1989 study the PATPs were undiluted because the researchers selected males reporting a single sexual exposure [7]. In the Hayes et al., 1995 study the authors estimated the number and duration of genital ulcer episodes and used Equation (2) to calculate PATPs both with and without an active genital ulcer present, therefore estimating an undiluted RR [8].

The remaining studies did not attempt to distinguish between sex acts with or without active genital ulcers. Rather, they periodically performed a genital examination in each follow-up visit, and asked the participant about the presence of genital ulcers *at any time* (the duration being unspecified) during the period since the preceding follow-up visit. The RR calculated was the ratio between the risk of having acquired HIV in the same period when people had GUD symptoms and the risk for people without them. Although they often used methods of regression analysis (such as Poisson) to calculate PATPs, different from our simpler equations above, the results of their calculations were diluted PATPs and RRs for GUD.

As the table shows, the undiluted PATPs and RRs were much higher than the diluted ones, corroborating the theoretical expectation.

### 3.2. A Critical Assessment of Current Estimates of the GUD Co-Factor Effect on HIV Transmission

The Boily et al., 2009 meta-analysis included a secondary analysis about HIV-1 PATPs with GUD and GUD risk ratios (Figure 3 in Ref. [16]). They arrived at a pooled estimate of PATP (index→exposed_GUD_) of 2.77% (0.51–14.98%), and a GUD risk ratio of 5.29 (1.43–19.58). For this secondary analysis they included five studies, which we mark with note 3 in our Table 1. We re-analyzed these papers [7,8,9,10,11], and the related papers that describe incidence of STIs, behavioural data, and other aspects of the same cohorts [12,13,24,25,26,27,28,29,30,31], and found some issues with the earlier meta-analysis.

First, we note that the Cameron et al., 1989 and the Gray et al., 2001 studies should not have been included in this set (which Boily et al. say is of index→exposed_GUD_ studies), because the latter certainly [10], and the former most likely [7], represented an index_GUD_→exposed situation (see also Section 1).

Second, the meta-analysis calculated a pooled index→exposed_GUD_ PATP for these five studies without distinguishing between studies that measured diluted and undiluted GUD-associated PATPs. This averaged out all estimates and the resulting estimates for PATPs and RRs, thereby obscuring the fact that the undiluted PATPs were much higher.

Several other meta-analyses [14,15,17] and many citing papers have stated that the GUD co-factor effect on HIV-1 PATP, both in index→exposed_GUD_ and in index_GUD_→exposed situations had RRs ranging between about 2 and 5. This is not wrong per se if we are talking about diluted RRs, as Table 1 shows, but such statements should come with a warning: RR can be much higher when one of the partners has an active genital ulcer ([7,8]; Table 1).

### 3.3. Reconciling Diluted and Undiluted Measurements

We can calculate diluted GUD-associated RRs and PATPs from studies that published the undiluted ones and vice versa. As an illustration, we perform this calculation based on the two studies that provided the most data to enable the assessment. Let us start with Hayes et al., 1995 [8]. The relevant quantitative data regarding frequency of sex, frequency of genital ulcers and other data were extracted from either Hayes et al., 1995 [8], or the primary paper reporting the original observations (Plummer et al., 1991 [24]). They calculated PATPs in a cohort of 117 HIV-1 seronegative CSWs exposed to clients [24]. Hayes et al., 1995 had data regarding duration of follow-up, frequency of sex, frequency of condom use, and HIV prevalence in the clients of these CSWs and, based on these data, they calculated and published the number of exposures, n, for both CSWs who acquired GUD and for those who did not. Of those who were GUD-free, 55% seroconverted to HIV-1 (*I* = 0.55), and they had an average of *n* = 246 exposures to HIV-1 infected clients. Hayes et al. used Equation (1), and calculated a baseline (without GUD) PATP of *P* = 0.0032. From those who had reported GUD, 72% seroconverted (*I_g_* = 0.72). The average number of exposures in this group was *n* = 249. Based on the duration of follow-up and the number of GUD episodes reported during follow-up, and assuming three alternatives for the average durations of ulcers (3 days, 1 week, 2 weeks) [8], we obtain the probability that each exposure overlapped with a genital ulcer, and from this the expected numbers of exposures with genital ulcers, *n_g_* = {2.84, 6.61, 13.23}, and without genital ulcers, *n*_0_ = *n* − *n_g_*. From Equation (3), we calculate the undiluted PATPs for the three ulcer duration assumptions, *P_g_* = {0.157, 0.0723, 0.0384}, and *R_g_* = *P_g_*/*P* = {49.0, 22.6, 12.0}, coinciding with the authors’ results [8].

We then calculate the diluted PATPs. We have the same *I_g_* = 0.72 and *n* = 249 exposures, all assumed to have the same GUD-associated co-factor effect. From Equation (4), we obtain:(5)$$P_{gdil}=1-{\left(1-I_{g}\right)}^{1/n}$$
which gives *P_gdil_* = 0.00510, and thus the risk ratio is *R_gdil_* = *P_gdil_*/*P* = 1.59. This is within the range of the diluted RRs displayed in Table 1.

As an example, in the opposite direction, we can calculate undiluted RRs and PATPs from the published diluted estimates presented by Gray et al., 2001 for rural Rakai, Uganda [10]. They calculated, using Poisson regression, a *P_gdil_* = 0.0041, and *P* = 0.0011. For couples in which the index had GUD during follow-up, mean follow-up was 19.0 months and mean number of sex acts per month was 7.16, leading to a mean *n* = 136.04. Applying Equation (4) we obtain *I_g_* = 0.42817. Another paper (Gray et al., 1999 [31]) covers the larger set of HIV-1 seronegative people from rural Rakai included in the Rakai randomized controlled trial (RCT) of STD treatment, of which the serodiscordant couples studied in Gray et al., 2001 [10] are a subset. It mentions that the mean reported duration of genital ulceration (potentially including more than one episode) was 1.2 (SE ± 0.3) months, and each follow-up period was 10 months [31]. There was evidence that underreporting of ulcers was low [31]. Thus, for the follow-up periods in which GUD was reported, genital ulcers were present in about 9–15% of time. If we assume that in the Gray et al., 2001 [10] smaller cohort of rural Rakai, the standards of GUD reporting were similar, *n_g_* would be *n* × {0.09, 0.15} = {12.244, 20.406} and *n*_0_ = *n* − *n_g_* = {123.80, 115.63}. Applying Equation (3), we obtain the undiluted *P_g_* = {0.03393, 0.02094}, and the risk ratios *P_g_*/*P* = {30.85, 19.04}. These RRs are similar to the published undiluted ones (Table 1).

This exercise does not replicate the exact conditions existing in Rakai, and we do not want to give the impression that it is accurate, or that the method used is the only one possible. Underreporting of ulcers may have happened, particularly of the shorter and less noticeable HSV-2 recurrences. The 9–15% genital ulcer time during follow-up that was seen in people reporting genital ulcers in the larger Rakai set (Gray et al., 1999 [31]) may not have been exactly replicated in the smaller subset studied by Gray et al., 2001 [10]. However, genital ulcer time in the latter study was unlikely to have been very different from the range found in the former study, thus likely leading to undiluted risk ratios comparable to the ones we calculate above. Our examples here show how the low diluted GUD-related PATPs naturally turn into high undiluted ones if we just introduce into the calculations the fact that genital ulcers were active in a small proportion of follow-up time. This proportion is the critical variable in this effect and will likely cause the above conclusions to hold even if different statistical methods are used.

## 4. Discussion and Conclusions

We have shown in this review that the highest estimates of GUD effect on HIV transmission (10–100-fold increased risk) are supported by the re-analysis of studies that yielded apparently lower estimates. Most studies that attempted to measure the co-factor effect of GUD in HIV-1 PATPs have calculated diluted forms of PATPs and related RRs, i.e., as if genital ulcers had been present all the time during follow-up. The RRs obtained were moderate (Table 1). This dilution is a form of exposure misclassification, as defined in epidemiology [32]. We must stress that the calculation of diluted forms of RR is not wrong per se, but may be misleading if the reader does not realize that it refers to a diluted RR. Only two studies so far have estimated undiluted GUD effects in HIV-1 PATP, one by selecting males exposed with just one contact with a CSW [7], and the other using Equation (2) (see Section 2) [8]. They calculated undiluted RRs in the range of 12–49 and 60–320 [7,8]. The very high latter estimate is based on a baseline PATP of 0.001 [8]. Had it been based on a baseline of 0.0035 (as calculated by Boily et al., 2009 [16]), the RR would still be in the range of 21–112 (Table 1). Meta-analyses have tended to mingle together diluted and undiluted estimates, did not attempt to differentiate between them, and calculated pooled estimates based on all, which was methodologically unsound. They usually calculated summary estimates of RRs similar to the lower diluted ones displayed in Table 1 [14,15,16,17]. These influential analyses have been echoed in many medical articles. This brings a practical problem: thousands of medics and people at risk, such as CSWs and others may not realize that having sex with an active genital ulcer present may pose a risk of HIV transmission between one and two orders of magnitude higher than currently established estimates. This is all the more important because the growing threat of antibiotic resistance in STI-causing bacteria [33] will likely increase genital ulcer frequency. Additionally, the implementation of UNAIDS 95-95-95 targets will cause HIV to become concentrated again in disadvantaged and high-risk groups, increasing the contribution of genital ulcers to the HIV epidemic.

An historic perspective is relevant in this Discussion. Several of the studies that found important co-factor effects of STIs and GUDs on HIV transmission were performed in the late 1980s and early 1990s [7,8,9,24]. Following that evidence, several randomized controlled trials (RCT) of STI treatment were designed and implemented, and the results published in the mid to late 1990s. The Mwanza trial was very successful, achieving a 38% reduction of HIV incidence [34]. Several other trials, including an early one done in Rakai, showed no significant effect (reviewed in [5,35]). This led to widespread skepticism about the value of mass STI treatment for HIV control, and by conjecture, about the importance of STI co-factor effects on HIV. However, several factors explain the discrepant results between the Mwanza and the other trials. It has been shown that the impact of STI co-factors on HIV might depend on the phase of an HIV epidemic [5,6,35]. HIV prevalence was low and growing fast in Mwanza (the earliest trial), which is associated with a stronger STI effect [4,5,6,35]; in contrast, the HIV epidemic was already generalized in Rakai and the other trials. STI prevalence was higher in Mwanza. A larger fraction of genital ulcers in Rakai were caused by HSV-2, rather than bacterial STIs. In Rakai there were widely spaced rounds of mass treatment, but between rounds, treatment was uncommon, while in Mwanza improved treatment was administered throughout the study period [5,35]. The other trials had other problems, including lack of statistical power and insufficient contrast between the treatment and control arms [35]. Therefore, from the failure to reduce HIV incidence seen in most trials, we cannot conclude that the STI co-factors on HIV are not important.

The diluted/undiluted distinction we develop here was pioneered by Hayes et al., 1995 [8] in their analysis of a previous study of CSWs exposed to GUD and HIV in Nairobi [24]. These calculations have limitations. Korenromp et al., 2001 pointed out that they are liable to several types of confounding [36]. For example, CSWs who contracted GUD, compared with those who did not contract it (i) may have had more sex acts; (ii) may have had more sex acts with clients with GUD who are more likely to transmit HIV; (iii) may have had more uncircumcised clients since GUD and lack of circumcision are correlated [37], and uncircumcised males have higher HIV prevalence. All three effects might contribute to a higher incidence of HIV in CSWs reporting GUD compared with those not reporting it, independent of the GUD co-factor effect in index→exposed_GUD_ PATP. Thus, Korenromp et al., 2001 [36] suggest that the Hayes et al., 1995 [8] estimates of the co-factor, and by implication, ours here, are likely to be biased upwards [8]. We note, however, that these arguments apply to both diluted and undiluted PATPs, and therefore do not affect the substantial difference between the two estimations, which is the main focus of our article.

In addition to the above referred confounding effects [36], another phenomenon that can blur the high co-factor effect of genital ulcers is that, after they heal, inflammation may persist in the area for weeks, as was demonstrated for HSV-2 [38]. This inflammation also likely increases the odds of HIV acquisition and transmission and may explain part of the diluted RRs found in studies (Table 1). While this would reduce the inferred undiluted co-factor effect of the ulcers themselves, the latter would still be considerably higher than the diluted one.

Still another source of bias that could lead to overestimation of undiluted GUD-associated PATPs and risk ratios would be if either the duration or the frequency of genital ulcers had been underreported in the original studies. This would lead to a higher real proportion of time with genital ulceration, and thus a higher real *n_g_* than was reported in the studies, and thus to lower undiluted *P_g_* and *R_g_*. Most of the empirical studies reviewed here were done at a time when ART was not available. Given the strong relationship between HIV viral load and infectivity [26], all PATPs we discuss here will be lower if the index individual is on ART. While the overall transmission of HIV will be lower in this setting, the amplifying effect of GUD might actually become more relevant for the continued transmission and persistence of the virus, when baseline transmissibility is very low. In addition, key populations such as CSWs, men who have sex with men (MSM), transgender individuals, and intravenous drug users (IVDU) tend to have both a far higher HIV prevalence than the general population (>20 times higher in the case of IVDU and MSM and ~10 times higher in the case of CSWs [39]) and a lower access to HIV testing and treatment, because of stigma and legal prosecution in many countries [39]. This means that these people will continue to form a pool of high HIV, high GUD, and low ART for years to come. In addition, other events, such as the current SARS-CoV-2 pandemic, may reduce ART uptake [40] and result in more HIV transmission, and hence, a heightened imperative in controlling GUD.

While the existence of a very high co-factor effect with active genital ulcers has previously been supported by only two studies [7,8] our results demonstrate that these estimates can actually be recovered by a detailed re-analysis of other studies that previously yielded lower estimates. The main conclusions of our analysis are that (i) the current paradigm of low GUD co-factor effects, with many papers claiming associated RRs around 2–5 is misleading because it reflects studies that measured diluted RRs; therefore (ii) all studies aimed at estimating GUD effects on HIV transmission should identify individual exposure events, and (iii) HIV prevention strategies should be based on the undiluted estimates of the GUD co-factor effect. These conclusions are supported by the finding of a very high GUD co-factor effect in all studies that measured it in its undiluted form [6,7]. The main limitation of these conclusions rests in the scarcity of such studies.

General HIV prevention measures, such as ART and condom use, have a large scope of application and are able to nearly stop transmission by suppressing the virus or imposing a physical barrier. Control of GUD should be most effective when directed to key populations such as CSWs and MSM and, in this regard is like pre-exposure prophylaxis (PreP). Both GUD control [34], PreP [41], and condom use [41] have been shown to reduce HIV incidence.

The very strong facilitating effect of active genital ulcers has practical relevance for the prevention of HIV transmission at the population level. We suggest revising HIV prevention guidelines to take into account a 10–100-fold increased risk of HIV transmission when a genital ulcer is present. STI treatment programs could be designed in a way that people at risk are observed very closely and with a high visit frequency, so that ulcers are observed and treated in time. Finally, guidelines could recommend to not have sex at all if one of the partners has a genital ulcer, perhaps not even with condoms, because the latter can break or slip. Both clinicians and people at risk should take home the message that having unprotected sex coinciding with active genital ulcers in either the index or the exposed partner may pose an unacceptable risk of HIV transmission.

## Figures and Tables

**Table 1 viruses-14-00538-t001:** The studies that estimated HIV-1 PATPs with GUD present or the risk ratio of GUD over baseline PATP.

Study	Direction	Sample	Location and Time	PATP with GUD ^1^	PATP without GUD or Overall	Risk Ratio of GUD	Type
GUD in index							
Cameron et al., 1989 [7] **^2^**^,**3**^	F→M (CSWs)	73 exposed men (to CSWs)	Nairobi, Kenya, 1986–87	Overall: 16.22% (3.24–29.2) Circumcised: 6.7% **^4^** Uncircumcised: 42.8% (12.7–73.0) **^4^**	0% (0.0–0.033)	160 (60–320) **^5^** 56 (21–112) **^6^**	Undiluted
Gray et al., 2001 [10] **^3^**	F→M M→F	174 serodiscor-dant couples	Rakai, Uganda, 1994–98	0.41% (0.0008–0.812)	0.11% (0.0073–0.144)	3.73 **^8^** Adjusted: 2.58 (1.03–5.69)	Diluted
Wawer et al., 2005 [13]	F→M M→F	235 serodiscor-dant couples	Rakai, Uganda, 1994–99	NA	0.12% (0.09–0.015)	2.19 (1.07–4.47) Adjusted: 2.04 (1.04–3.99)	Diluted
Hughes et al., 2012 [18]	F→M M→F	3297 serodiscor-dant couples	14 sites, Eastern and Southern Africa	NA	NA	F→M: 0.32 (0.044–2.30) M→F: 0.84 (0.20–3.51)	Diluted
GUD in exposed							
Hayes et al., 1995 [8] **^3^**^,**8**^	M→F (CSWs)	124 exposed CSWs	Nairobi, Kenya, 1985–87	7.36% (3.84–15.68)	0.320% (0.199–0.441)	23 (12–49) 21.0 (11.0–44.8) **^6^**	Undiluted
Mastro et al., 1994 [9] **^3^**	F→M (CSWs)	1115 exposed men (military conscripts)	Northern Thailand, 1988–91	4.1% (3.1–5.4)	2.0% (1.3–3.1)	2.9 (1.7–5.1)	Diluted
Corey et al., 2004 [12]	F→M M→F	174 serodiscor-dant couples	Rakai, Uganda, 1994–98	0.31% (HSV-2+)	0.19% (HSV-2+) 0.11% (all)	1.63 **^7^** 2.82 **^7^**	Diluted
Baeten et al., 2005 [11] **^3^**^,**9**^	F→M (Some CSWs)	745 exposed men (truck drivers)	Mombasa, Kenya, 1993–97	0.730% (0.146–1.314)	<0.63%	<1.16 **^7^**	Diluted
Hughes et al., 2012 [18]	F→M M→F	3297 serodiscor-dant couples	14 sites, Eastern and Southern Africa	NA	NA	F→M: 2.04 (0.72–5.77) M→F: 3.60 (1.52–8.49)	Diluted

^**1**^ “with GUD” means GUD was present at any time during the follow-up for studies which calculated diluted risk ratios and PATPs; **^2^** Males exposed with a single encounter with a CSW; **^3^** These studies were included in the Boily et al., 2009 [16] secondary analysis about GUD effects on HIV PATPs. **^4^** These PATPs were calculated by survivorship analysis [7]; **^5^** Calculated using a baseline PATP of 0.1% [8]; **^6^** Calculated by us based on a baseline PATP of 0.35% [16]; **^7^** Risk ratio *P_gdil_*/*P* calculated by us; **^8^** The authors estimated the proportion of acts in which CSWs had active genital ulcers; **^9^** Includes exposure of males to both CSWs and other women; the PATP without GUD is not provided but the overall PATP is 0.63%.

## Data Availability

Not applicable.

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
