# Peer review of "The Impact of Genital Ulcers on HIV Transmission Has Been Underestimated—A Critical Review"

_viruses, 2022, doi:10.3390/v14030538_

Round 1

Reviewer 1 Report

The authors did answer to my questions and supplement the discussion regarding the sources of bias. The main limitation is still the scarcity of the number of studies publishing undiluted RR and available datasets, limiting the generalization of the results and yielding to a large uncertainty in the increased risk.

This limitation could be added to the final conclusion.

Author Response

We respond to the Reviewer in the uploaded file.

Reviewer 2 Report

HIV infection and sexually transmitted diseases remain major public health concerns that affect people living in settings with limited resources, including limited access to health care and medications, significantly harder than populations with comprehensive health care coverage and access to medications. The meta analysis and review by Sousa and colleagues provides support for the notion that the role of diseases associated with genital ulcers or infections causing them in HIV infections has been underestimated. Moreover, the authors suggest an update of HIV care information and recommendations.
While the studies reviewed and subjected to the meta analysis are limited in number as wells as geographically and in terms of the time periods they cover, the authors acknowledge those issues. The data also suggests that further studies are needed both analytically and interventionally to address the problem. It would be helpful, if the authors could include in table 1 the number of subjects of each study and also summarize other demographic information. That could increase the impact of the manuscript and strengthen the argument for further studies with regard to numbers of people who should be enrolled and what geographic location have been covered or need to be studied in the future in order generate additional statistically meaningful data.

Author Response

(The authors gave the same response as above.)
